# Bilateral Remote Ischaemic Conditioning in Children (BRICC) trial: protocol for a two-centre, double-blind, randomised controlled trial in young children undergoing cardiac surgery

Nigel E Drury [1,2] Rehana Bi,[1,3] Rebecca L Woolley,[4,5] John Stickley,[1] Kevin P Morris,[3,5] James Montgomerie,[6] Carin van Doorn,[7] Warwick B Dunn,[8,9] Melanie Madhani,[2] Natalie J Ives,[4,5] Paulus Kirchhof [2,10] Timothy J Jones[1,2]

**Correspondence to**
Mr Nigel E Drury;
n.e.drury@bham.ac.uk

## ABSTRACT

**Introduction** Myocardial protection against ischaemic-reperfusion injury is a key determinant of heart function and outcome following cardiac surgery in children. However, with current strategies, myocardial injury occurs routinely following aortic cross-clamping, as demonstrated by the ubiquitous rise in circulating troponin. Remote ischaemic preconditioning, the application of brief, non-lethal cycles of ischaemia and reperfusion to a distant organ or tissue, is a simple, low-risk and readily available technique which may improve myocardial protection. The Bilateral Remote Ischaemic Conditioning in Children (BRICC) trial will assess whether remote ischaemic preconditioning, applied to both lower limbs immediately prior to surgery, reduces myocardial injury in cyanotic and acyanotic young children.

**Methods and analysis** The BRICC trial is a two-centre, double-blind, randomised controlled trial recruiting up to 120 young children (age 3 months to 3 years) undergoing primary repair of tetralogy of Fallot or surgical closure of an isolated ventricular septal defect. Participants will be randomised in a 1:1 ratio to either bilateral remote ischaemic preconditioning (3×5 min cycles) or sham immediately prior to surgery, with follow-up until discharge from hospital or 30 days, whichever is sooner. The primary outcome is reduction in area under the time-concentration curve for high-sensitivity (hs) troponin-T release in the first 24 hours after aortic cross-clamp release. Secondary outcome measures include peak hs-troponin-T, vasoactive inotrope score, arterial lactate and central venous oxygen saturations in the first 12 hours, and lengths of stay in the paediatric intensive care unit and the hospital.

**Ethics and dissemination** The trial was approved by the West Midlands-Solihull National Health Service Research Ethics Committee (16/WM/0309) on 5 August 2016. Findings will be disseminated to the academic community through peer-reviewed publications and presentation at national and international meetings. Parents will be informed of the results through a newsletter in conjunction with a local charity.

**Trial registration number** ISRCTN12923441.

## Strengths and limitations of this study

► This is the first randomised controlled trial to evaluate the efficacy of bilateral remote ischaemic preconditioning, applied simultaneously to both lower limbs to provide a more intense stimulus in young patients undergoing surgery.

► It is also the first multicentre cardiac surgical trial in children in the UK.

► We will exclude neonates, in whom preconditioning may be harmful, and avoid the use of propofol anaesthesia, which is thought to interfere with the preconditioning pathway.

► A potential limitation is if exposure to cyanosis in those with tetralogy of Fallot has already had a preconditioning effect, this could attenuate the effect of the intervention.

► The effect of the intervention may also be concealed if right ventricular incision, muscle resection or outflow tract stent removal significantly increases troponin release in patients with tetralogy of Fallot above that associated with ischaemia reperfusion.

## INTRODUCTION
### Myocardial protection

During most surgery for congenital heart disease, it is necessary to stop the heart, allowing access to a still and bloodless field to enable repair of intracardiac defects. Cardioplegia and hypothermia have been fundamental to arresting the heart and protecting against ischaemia-reperfusion (IR) injury during surgery for over 40 years and are used in approximately 3500 cardiac surgical operations in children in the UK and Ireland each year.[1] However, the developing myocardium exhibits marked differences in metabolism from the adult heart[2] and as current techniques for cardioprotection were developed in adults, they may not be optimal for

young children.[3 4] Myocardial injury still occurs routinely following aortic cross-clamping in children,[2 5] with IR leading to a degree of contractile impairment which may manifest as low cardiac output and require inotropic support in the early postoperative period. This is a major cause of morbidity and death in the early postoperative period[6 7] and children with preoperative cyanosis are more vulnerable to the effects of IR than acyanotic children.[8 9] Postoperative elevation of circulating troponin is a biomarker of myocardial injury and has been shown to strongly correlate with clinical outcomes including level of inotropic support, duration of ventilation, ventricular dysfunction and early death[5 10]; consequently, it is the most common primary outcome measure in clinical trials of cardioprotection in children.[11] Myocardial protection therefore is a key determinant of heart function and outcome following cardiac surgery.

### Remote ischaemic preconditioning

Remote ischaemic preconditioning (RIPC) involves the application of brief, non-lethal cycles of ischaemia and reperfusion to a distant organ or tissue, such as a limb, to induce protection against subsequent myocardial IR injury.[12] There are thought to be two phases of cardioprotection: a first window with an immediate effect lasting several hours, and a second window which appears around 12–24 hours and lasts for 48–72 hours.[13] The stimulus has traditionally been applied to the upper arm (adults) or thigh (children) using a blood pressure cuff inflated to above systolic pressure.[14] The promise of this simple, low-risk, inexpensive and readily available technique as an adjunct to current methods for myocardial protection has prompted numerous trials in adults[15–20] and children[21–28] but with mixed results. A meta-analysis suggested that RIPC reduces myocardial injury in both adult and paediatric cardiac surgery,[29] but subsequently two large multicentre trials in adults failed to show benefit in either composite cardiovascular endpoints or troponin release[19 20]; both have been criticised for using propofol anaesthesia after it had been suggested to interfere with the preconditioning pathway.[30 31]

Cheung *et al* first demonstrated reductions in troponin release and perioperative inotropic requirements in a heterogeneous cohort of children, most of whom had either tetralogy of Fallot or ventricular septal defect (VSD).[21] Several studies have found improved myocardial protection in infants and young children undergoing tetralogy of Fallot repair[28] or VSD closure,[22 23] while others have found no benefit[24 25] and suggested that preoperative cyanosis may have already up-regulated pro-survival pathways.[25] The only trial in cyanosed neonates found no benefit, citing young age, myocardial immaturity and chronic hypoxaemia as potential conflicting factors[26]; animal models have also suggested that preconditioning may have no effect[32] or even be harmful[33] to the immature myocardium. To date, no clinical trials have compared the effects of RIPC in patients with or without

chronic cyanosis and its impact on preconditioning remains uncertain.[34]

In the largest paediatric trial to date, McCrindle *et al* found no benefit in clinical outcomes, physiological markers or subgroup analyses in a mixed cohort of 299 children[27] and proposed that better than expected outcomes in the control group, heterogeneity of underlying conditions and use of propofol may have affected their findings. Failure to elicit a stimulus may also have been a key factor; manual inflation of the cuff to just 15 mm Hg above systolic pressure may have led to periods of subclinical reperfusion and abolition of any preconditioning response. A recent meta-analysis in children determined that RIPC has a cardioprotective effect, with reduced troponin release, lower inotrope scores and reduced paediatric intensive care unit (PICU) stay following surgery[35] but was unable to include the largest trial in most analyses due to a lack of suitable published data.

### Rationale

In this trial, we will test whether in young children undergoing primary repair of tetralogy of Fallot or closure of an isolated VSD, the two most common congenital heart defects requiring surgery,[1] adequately delivered RIPC, compared with sham inflation–deflation cycles, improves myocardial protection. The design will enable evaluation of the effects of RIPC in children with and without preoperative cyanosis[34]; most patients with tetralogy of Fallot have chronic hypoxaemia while those with a VSD are not usually cyanotic and both groups undergo surgery at a similar age. We will use a more intensive two-cuff technique,[18] applying a concurrent stimulus to both lower limbs to compensate for the lower skeletal muscle mass in young children. We will address methodological concerns by using a pressure-controlled tourniquet system set to at least 50 mm Hg above systolic pressure,[27] avoiding propofol anaesthesia,[30 31] and not enrolling neonates or other infants less than 3 months old.[26] We will only seek to exploit the first window of preconditioning, performing the intervention under general anaesthesia prior to sternotomy, as the second window would require RIPC at least 12 hours prior to surgery[13] which may be logistically challenging, distressing to the child and their parents, and lead to incomplete intervention or withdrawal. Finally, this trial will be the first multicentre cardiac surgical trial in children in the UK[36] and act as a primer for the development of a network for the design and conduct of multicentre phase III trials in paediatric cardiac surgery in the UK and Ireland.

## METHODS AND ANALYSIS
### Design

The Bilateral Remote Ischaemic Conditioning in Children (BRICC) trial is a two-centre, double-blind, parallel arm, randomised controlled trial (RCT) to investigate the effects of RIPC and the impact of cyanosis on myocardial

protection in young children undergoing elective cardiac surgery. It will be conducted through the Birmingham Clinical Trials Unit (BCTU), a UK Clinical Research Collaboration-registered clinical trials unit with expertise in surgical and paediatric trials.

## Inclusion and exclusion criteria
### Inclusion criteria
All infants and young children, aged 3 months to 3 years at the time of surgery, undergoing either primary repair of tetralogy of Fallot or surgical closure of a VSD, with or without concomitant atrial septal defect (ASD) closure or pulmonary artery repair/augmentation, at Birmingham Children's Hospital or Leeds Children's Hospital will be included. Only patients with the most common form of tetralogy of Fallot will be included; variants such as absent pulmonary valve syndrome, pulmonary atresia with major aortopulmonary collateral arteries or with an atrioventricular septal defect will not be included.

### Exclusion criteria
The following children will be excluded from the study:
▶ Those requiring an additional procedure (other than ASD closure or pulmonary artery repair/augmentation) at the time of primary repair for example, mitral repair, aortic arch repair.
▶ Those with significant airway or parenchymal lung disease, bleeding disorder or a recent ischaemic event.
▶ Those who have undergone a previous cardiac surgical procedure with cardioplegic arrest.
▶ Those presenting in a critical condition and requiring emergency surgery.
▶ Those for whom the parents are unwilling or unable to give informed consent.

## Recruitment
Both tetralogy of Fallot and VSD are congenital heart defects that usually present with gradual onset of symptoms such as failure to thrive, difficulty feeding, dyspnoea or cyanosis. The referral pathway is therefore predictable with most children undergoing elective surgery following a period of medical therapy to allow them to grow; some children may require a palliative procedure prior to repair, notably right ventricular outflow tract (RVOT) stenting for cyanosis in tetralogy of Fallot[37] or pulmonary artery banding to reduce pulmonary overcirculation with an unrestrictive VSD. All eligible patients will be identified from the multidisciplinary team meeting, surgical clinics or waiting lists by the principal investigators at each site, and their parents approached to ascertain interest in the trial. They will be provided with a parent/guardian information sheet (online supplemental appendix A and B) either in the clinic/ward or sent in the post and given at least 24 hours to consider their child's participation and ask questions. Written informed consent will be obtained by a consultant surgeon prior to enrolment (online supplemental files C and D). The participant pathway through the trial is shown in figure 1.

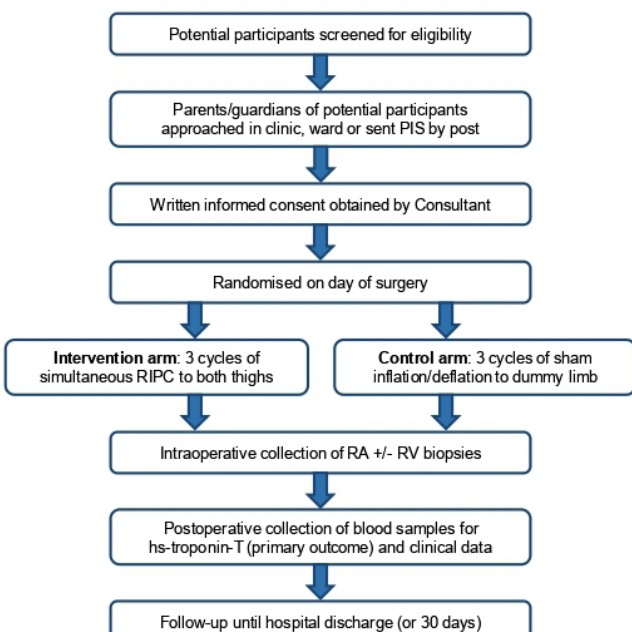

**Figure 1** Participant pathway from screening to end of follow-up. PIS, parent information sheet; RA, right atrium; RIPC, remote ischaemic preconditioning; RV, right ventricle.

## Randomisation and blinding
On the day of surgery, participants will be randomised in a 1:1 ratio to either RIPC or sham procedure using a secure online randomisation system, with a minimisation algorithm incorporating the following factors:
▶ Congenital heart defect: tetralogy of Fallot or VSD.
▶ Presence of an RVOT stent in patients with tetralogy of Fallot.
▶ Surgical centre: Birmingham or Leeds.

To avoid any possibility of the allocation becoming predictable, a random element will be included in the algorithm. If online randomisation is unavailable, a telephone helpline with emergency paper randomisation will be used. An independent healthcare professional, trained and competent in delivering the trial intervention, will perform the randomisation and administer the allocated treatment according to a standard operating procedure; the research nurse, surgical, anaesthetic, perfusion and PICU teams involved in the child's care will remain blinded to group allocation throughout the trial.

## Treatment arms
### Intervention arm
After induction of anaesthesia but prior to sternotomy, the treatment group will receive RIPC induced by three cycles of 5-minute ischaemia and 5-minute reperfusion.[38] Ischaemia will be induced simultaneously in both lower limbs using the PTSii system (Delfi Medical Innovations, Vancouver, Canada), a state-of-the-art digital tourniquet with precise control of occlusion pressure. Age-appropriate PediFit cuffs, with contour limb protection sleeves, will be placed around both thighs and inflated to at least 50 mm Hg above systolic pressure

measured in real-time via the arterial line during the ischaemia phase of each cycle. If one lower limb is unavailable, for example, required for vascular access during the intervention period, one cuff may be placed on the upper arm instead. In addition, a dummy limb will be placed between the patient's legs to maintain blinding (see the Control arm section). Continual loss of arterial flow will be confirmed by distal pulse oximetry during each limb occlusion cycle, visible only to the person applying the intervention[25] ; if the distal trace is not rapidly lost, the cuff will be tightened or the inflation pressure increased to achieve arterial occlusion. If pulse oximetry is not available, a clinical assessment will be made to determine whether there is loss of arterial flow (decreased lower limb temperature to touch, marked prolongation of capillary refill time) and reperfusion (increased lower limb temperature±blushing) during each cycle. Once the intervention has begun, each cuff must be kept on the same limb to ensure repeated doses of IR to the same muscle mass. Blinding will be maintained by covering the child with a surgical drape from above the nipples downwards including all four limbs throughout the period of cuff application, intervention and removal.

### Control arm

Contour limb protection sleeves will be placed around both thighs but the PediFit cuffs will be attached to the dummy limb (43×300 mm polyethylene tubing) placed between the patient's legs. Three sham inflation-deflation cycles will be performed using the PTSii system. Pulse oximetry monitoring will be reviewed by the person applying the intervention only, but no loss of trace will be observed during the cycles. As above, the child will be covered with a surgical drape to maintain blinding before, during and after the sham intervention.

Adherence to treatment will be defined as receiving the allocated treatment, and in the intervention arm, with loss of arterial flow (pulse oximetry or clinical assessment, if required) during each period of limb ischaemia.

### Common aspects of care
#### Anaesthesia

Anaesthesia will be conducted at the discretion of the consultant anaesthetist and involve a balanced technique using volatile and intravenous anaesthesia and adjuncts, opioid pain relief and muscle relaxants, within the limits of the protocol. Propofol will not be used for induction or maintenance of anaesthesia; isoflurane will be the preferred volatile anaesthetic agent and end-tidal partial pressure will be recorded at the end of RIPC administration. Phenylepherine will be used for vasoconstriction, as required. Routine monitoring will include continuous invasive arterial and central venous pressures, other cardiac output variables, urine output, blood gas analysis and near-patient clotting profile.[39] Systemic anticoagulation will be achieved with heparin prior to institution of cardiopulmonary bypass (CPB) and reversed with protamine after the termination of CPB.

### Surgery and perfusion

Repair of the congenital heart defect(s) will be performed following best clinical practice. After transfer to the operating room, the surgical checklist will be completed, the patient prepped and draped, and the chest opened through a median sternotomy. Standardised CPB will be established between the vena cavae and the ascending aorta with moderate hypothermia. An aortic cross-clamp will be applied to the proximal ascending aorta with intermittent antegrade cold cardioplegia given via the aortic root for myocardial protection; patients undergoing VSD closure will usually receive a single dose, while those with tetralogy of Fallot will typically require an additional dose. Removal of the aortic cross-clamp with myocardial reperfusion will be considered as time zero for the recording of postoperative events. Following completion of the repair and rewarming, CPB will be weaned and discontinued. In the event of difficulty separating from bypass or marked haemodynamic instability, subjective and objective measures of ventricular function will be obtained, and inotropic support instituted at the discretion of the blinded operating team. Once haemodynamic stability and haemostasis have been achieved, the chest will be closed at the discretion of the surgical team and the patient transferred to the PICU. Standard postoperative care will proceed with anticipated removal of the arterial line at 12 hours following surgery, removal of the central line at 24 hours and transfer to the ward once routine PICU discharge criteria have been met. All decisions regarding escalation of therapy will be made by the blinded clinical team responsible for the care of the child without influence from the researchers.

### Trial investigations

The schedule for the intervention and collection of outcome data, blood and tissue samples is shown in table 1.

#### Data collection

Clinical data will be collected by the research nurse before, during and after surgery. Inotrope usage in the first 12 hours will be used to generate a vasoactive inotrope score (µg/kg/min).[40 41] Arterial lactate and central venous oxygen saturations will be recorded prior to surgery and at 3, 6, 9 and 12 hours. Length of stay in PICU (hours) and hospital (days) following surgery will be documented. Preoperative haematocrit and resting oxygen saturations in air will be used as markers of the degree of exposure to cyanosis. In Birmingham only, cardiac output will be measured over the first 12 hours following reperfusion using ICON (Osypka Medical, Berlin, Germany), a non-invasive technique for electrical velocimetry which has been validated in young children.[42–44]

#### Blood samples

Blood will be drawn from indwelling arterial or central venous lines at baseline (after induction of anaesthesia but prior to sternotomy) and at 3, 6, 12 and 24 hours

**Table 1** Schedule of events: intervention, outcome data, blood and tissue samples

| | Preoperative | Pre-sternotomy | Intraoperative | | | On PICU admission | Time since aortic cross-clamp removal (hours) | | | | | Daily until discharge | Hospital discharge |
| | | | Onset ischaemia | During ischaemia | Late ischaemia | | 3 | 6 | 9 | 12 | 24 | | |
|---|---|---|---|---|---|---|---|---|---|---|---|---|---|
| Screening and consent | x | | | | | | | | | | | | |
| Randomisation | x | | | | | | | | | | | | |
| Clinical baseline data | x | | | | | | | | | | | | |
| RIPC or sham intervention | | x | | | | | | | | | | | |
| Blood for high-sensitivity troponin–T | | x | | | | | x | x | x | x | x | | |
| Arterial/venous blood gases | | x | | | | | x | x | x | | | | |
| Right atrium biopsies | | | x | | x | | | | | | | | |
| Right ventricle biopsies | | | | x | | | | | | | | | |
| Clinical outcome data | | | | | | x | | | | x | x | x | x |
| Cardiac index (BCH only) | | | | | | x | x | x | x | | | | |

BCH, Birmingham Children's Hospital; PICU, paediatric intensive care unit; RIPC, remote ischaemic preconditioning.

after reperfusion. Plasma samples for high-sensitivity (hs) troponin-T (Elecsys Tn-T HS, Roche, Basel, Switzerland) will be collected in paediatric lithium heparin tubes, centrifuged, split into two aliquots and stored at −80°C in remotely monitored freezers at each site until transfer for analysis at one of two core labs (Sandwell General Hospital, Birmingham or Russells Hall Hospital, Dudley, UK). Samples will be analysed in batches approximately every 8 months so that data on the primary outcome will be available to the Data Monitoring Committee prior to each meeting.

### Tissue samples
In Birmingham only, myocardial biopsies will be obtained for a metabolic substudy. Right atrial samples will be taken soon after aortic cross-clamping (onset ischaemia) and just before its release (late ischaemia) to assess metabolic changes in the myocardium during the period of ischaemia. In a subset of patients with tetralogy of Fallot, several samples of hypertrophic septoparietal trabeculae of the right ventricular infundibulum will be obtained at various points during ischaemia, whenever routinely resected. Specimens will be briefly washed in saline, promptly snap-frozen in liquid nitrogen and stored at −80°C until transfer to the Phenome Centre Birmingham for metabolic phenotyping. Analysis of these samples is exploratory and will follow a separate analytical plan (see the Substudies section).

### Outcome measures and follow-up
#### Primary outcome:
Reduction in area under the time-concentration curve (AUC) for hs-troponin-T release in the first 24 hours after aortic cross-clamp release (reperfusion) as a marker of myocardial injury.

#### Secondary outcomes
► Peak hs-troponin-T in the first 12 hours.
► Total vasoactive inotrope score in the first 12 hours.
► Arterial lactate and central venous oxygen saturations in the first 12 hours.
► Length of postoperative stay in the PICU.
► Length of postoperative stay in the hospital.

#### Exploratory outcome
Cardiac index in the first 12 hours measured using ICON (Birmingham only).

#### Follow-up
Until discharge from hospital or 30 days, whichever is sooner.

### Analysis
#### Sample size
It is hypothesised that RIPC will reduce the AUC for hs-troponin-T release in the first 24 hours compared with controls, but that exposure to hypoxaemia may impact on this reduction. The sample size proposed here will be sufficient to detect a 35% reduction in

postoperative troponin release, assuming a mean release of 350 µg/L/hour in the control group compared with 228 µg/L/hour in the RIPC group (extrapolated from the similarly mixed cohort of hypoxic and non-hypoxic children in Toronto[21]), with a variability of 220 µg/L/hour.[24] A sample size of at least 52 children per treatment group is needed to have a power of 80% and a significance level of 0.05 (two-sided). We therefore will recruit at least 104 children (up to 120 children to allow for dropouts) randomised in a 1:1 ratio between RIPC and control.

### Expected recruitment rate

The paediatric cardiac surgery units in Birmingham and Leeds are ideally placed to conduct clinical trials. Over the preceding 3 years, 99–135 children per annum have undergone surgical repair of either tetralogy of Fallot (mean 50) or VSD (mean 69) across the two sites.[1] The only previous interventional trial in cardiac surgery at Birmingham Children's Hospital recruited 22 (79%) of the 28 patients approached.[45] None of the other UK paediatric cardiac surgery RCTs have reported recruitment rates[36] but our predictions are comparable to those obtained from similar trials in North America which ranged from 62% to 84%.[27 36 46] We will maintain a screening log to document exclusions and reasons given by parents who decline to participate; this will be available to the Trial Management Committee who will monitor recruitment targets and advise on any changes to the protocol.

### Statistical analysis

Analysis of the main outcome measures will be performed according to the intention-to-treat principle and any non-adherence to the allocated group documented. The primary analysis will assess whether RIPC reduces AUC for troponin release in the first 24 hours compared with control. The primary outcome measure will be calculated using the trapezoidal method and presented as an adjusted mean difference between groups along with the 95% CI estimated using a linear regression model (adjusting for the minimisation variables and baseline troponin). For the secondary outcomes, continuous data items (eg, peak troponin) will also be analysed using a linear regression model. Continuous outcomes measured across more than three time points (eg, arterial lactate and central venous oxygen saturations) will be analysed using mixed effect repeated measures models. Time-to-event data outcomes will be analysed using a Cox regression model. Test of interactions will be employed to assess whether there is evidence that the treatment effect differs between cyanotic and acyanotic patients. P values will be reported from two-sided tests at the 5% significance level. A detailed statistical analysis plan is under development and will be approved prior to database lock. The chief investigator and trial statisticians will have access to the final trial dataset.

## Monitoring

### Assessment and management of risk

No adverse events directly attributed to the application of a tourniquet cuff during RIPC were identified in a meta-analysis of 1762 adults and children undergoing cardiac surgery in 25 trials[29] nor in any of the notable trials published since.[19 20 27 28] Risk to participants therefore is deemed to be minimal and the trial is categorised as type A: no higher than the risk of standard medical care. In the event of concern, parents will be signposted to their cardiac specialist nurse, their general practitioner or the hospital Patient Advice Liaison Service, as appropriate.

### Trial Management Committee

The trial will be overseen by a committee meeting approximately every 4 months during the trial. It will comprise clinicians, trialists and scientists involved in the set-up and running of the trial including representation from both trial sites. During recruitment, the protocol may be reviewed considering achievement of recruitment targets, evidence from new publications and feedback from parents approached for the trial; ethical approval for amendments to the protocol will be sought, as required.

### Data Monitoring Committee

An independent Data Monitoring Committee will meet approximately every 8 months during recruitment to review efficacy and safety data, according to a predefined charter (online supplemental appendix E). Members are an academic consultant cardiac surgeon as chair, a consultant in paediatric cardiac intensive care and a statistician. Analysis of hs-troponin-T for the primary outcome will be performed in batches prior to each meeting and all unblinded safety and efficacy data made available to the committee.

### Safety reporting

Adverse events will be recorded and reported in accordance with the sponsor's Code of Practice for Research. Participants in the study are undergoing open heart surgery and therefore adverse events are anticipated. The following serious adverse events will be reviewed by the chief investigator and reported to the sponsor within 48 hours of identification: death; requirement for extracorporeal life support; evidence of a major neurological event; and need for further surgery in the early postoperative period.

### Data collection and management

All data will be entered onto the BRICC trial database, a password-protected electronic database held on secure University of Birmingham servers for trial data with access limited to BCTU members of staff working on the trial. All paper case report forms will be stored securely in the research offices at Birmingham Women's and Children's National Health Service (NHS) Foundation Trust and Leeds Teaching Hospitals NHS Trust. Data will be semi-anonymised by removing non-essential potentially identifiable patient information; blood and tissue samples will

be labelled with the unique trial ID number, date and time of collection. Adherence to trial processes will be audited by the independent Clinical Research Compliance team at the University of Birmingham.

## Substudies
### Metabolic phenotyping

No study in children has previously examined the impact of RIPC on myocardial metabolism or its interaction with chronic hypoxaemia. Therefore, building on metabolic phenotyping in animal models of IR injury,[47] we will analyse intraoperative biopsies to identify changes in myocardial metabolic pathways that occur during ischaemia. In brief, tissue extracts will be analysed using ultra high-performance liquid chromatography (UHPLC)– mass spectrometry in two independent discovery and validation phases. Two complementary assays will be applied, (1) hydrophilic interaction liquid chromatography (HILIC) assay to study water-soluble metabolites, including those present in glycolysis and the tricarboxylic acid cycle, and (2) $C_{18}$ reversed-phase assay to determine changes in lipids during ischaemia.[48] The eluents from UHPLC columns will be introduced directly into an electrospray Q Exactive Mass Spectrometer (Thermo Scientific, UK) and data acquired in the $m/z$ range 70–1000. The impact of RIPC on metabolism and how any changes may be attenuated by preoperative cyanosis will be assessed through robust statistical analysis using correction for multiple testing and pathway enrichment analysis.

### Qualitative

We will explore parents' perspectives on decision-making about their child's participation in a clinical trial as part of their elective cardiac surgery. Parents of children approached to participate in the trial, both consenters and decliners, will be contacted following surgery and asked to participate in semistructured interviews which, with written informed consent, will be digitally audio-recorded, intelligently transcribed and thematically analysed. The findings will enhance our understanding of the factors that influence parents' decision-making and be used to inform the design and conduct of future trials. The BRICC trial is a suitable vehicle for this substudy as the intervention presents minimal risk, the surgery is performed electively and the operations included have a low predicted mortality (STAT categories 1–2).[49]

### Patient and public involvement

Patient and public involvement (PPI) has been a central component in the development, conduct and planned reporting of this trial since its inception. Parents of children who had previously undergone cardiac surgery at Birmingham Children's Hospital were contacted through *Young at Heart*, the local children's heart charity. Four parents reviewed the draft parent information sheet and consent form for the trial, making suggestions to improve clarity and readability for a lay audience, which were

incorporated into the final versions. The parent information sheets, consent forms and protocol for the qualitative substudy were also reviewed by the Young Person's Steering Group in the West Midlands. The outcomes of the trial will be communicated by individual parent feedback and a charity newsletter, both of which will be produced in collaboration with the charity and parents. Early user involvement was funded by a bursary from the National Institute for Health Research (NIHR) Research Design Service West Midlands and all PPI was costed using the INVOLVE Calculator according to the NIHR's Budgeting for Involvement.[50]

## ETHICS AND DISSEMINATION

This clinical trial was approved by the West Midlands-Solihull NHS Research Ethics Committee (16/WM/0309) on 5 August 2016 and the NHS Health Research Authority (200876) on 19 August 2016. It is sponsored by the University of Birmingham (RG_14-025, email: research-governance@contacts.bham.ac.uk, telephone:+44 (0) 121 415 8011), registered on the NIHR Clinical Research Network portfolio (32330), and approved by the NHS Research and Development departments at Birmingham Children's Hospital (1845) and Leeds Children's Hospital (PA17/67348). Regulatory approval from the Medicines and Healthcare products Regulatory Agency was not required as this trial is not a clinical trial of an investigational medicinal product (CTIMP). The first patient was randomised on 24 October 2016 and recruitment is currently ongoing.

### Changes to the protocol since original ethical approval

Since the original ethical approval, four substantial amendments to the protocol have been sought and approved with the following significant changes:

► Add 'with/without concomitant pulmonary artery repair/augmentation' to the inclusion criteria, to allow inclusion of those with pulmonary artery disease within the spectrum of tetralogy of Fallot and those with VSD who had previous pulmonary artery banding (December 2016).
► Add Leeds Children's Hospital as the second site and extend the duration of recruitment (February 2018).
► Remove 'known major chromosomal defect' as an exclusion criterion; although originally included as per previous paediatric trials,[21 27] following discussion with Professor Andrew Redington (Cincinnati, Ohio, USA), principal investigator of these trials, it became clear that there was no biological reason relating to RIPC to exclude these patients (February 2018).
► Add Russells Hall Hospital, Dudley as a second core laboratory to maintain internal validity, as Sandwell General Hospital, Birmingham changed their troponin analysis platform during the trial (November 2019).

### Dissemination plan

The findings of the clinical trial and substudies will be submitted for presentation at national and international

meetings and manuscripts prepared for submission to leading journals. The authorship of the final trial report will include all members of the trial management committee and named collaborators.

Parents of children participating in the trial will be informed of the results in writing once data analysis is complete. The local charity *Young at Heart* will also report the outcomes in their newsletter to reach a wider audience of those affected by congenital heart disease. PPI collaborators will be invited to participate in producing both the parent feedback and charity newsletter.

The first author is chief investigator of the trial and takes responsibility for the integrity of this protocol report, which adheres to the Standard Protocol Items: Recommendations for Interventional Trials recommendations.[51] All authors have read and agree to the manuscript as written.

**Author affiliations**
$^1$Paediatric Cardiac Surgery, Birmingham Children's Hospital, Birmingham, UK
$^2$Institute of Cardiovascular Sciences, University of Birmingham, Birmingham, UK
$^3$Paediatric Intensive Care, Birmingham Children's Hospital, Birmingham, UK
$^4$Birmingham Clinical Trials Unit, University of Birmingham, Birmingham, UK
$^5$Institute of Applied Health Research, University of Birmingham, Birmingham, UK
$^6$Paediatric Cardiac Anaesthesia, Birmingham Children's Hospital, Birmingham, UK
$^7$Congenital Cardiac Surgery, Leeds Teaching Hospitals NHS Trust, Leeds, UK
$^8$School of Biosciences, University of Birmingham, Birmingham, UK
$^9$Phenome Centre Birmingham, University of Birmingham, Birmingham, UK
$^{10}$Cardiology, University Heart and Vascular Center, UKE, Hamburg, Germany

**Acknowledgements** We are grateful to the members of the independent Data Monitoring Committee for their guidance and oversight throughout the trial: Professor Gavin J Murphy (Chair), University of Leicester, UK; Dr Katherine L Brown, Great Ormond Street Hospital, London, UK; and Dr Peter Nightingale (Statistician), Queen Elizabeth Hospital Birmingham, UK. We thank Dr John V Pappachan (Southampton, UK) for guidance on the delivery of RIPC, Professor Andrew N Redington (Cincinnati, Ohio, USA) for advice on RIPC and the exclusion criteria, and Professor Peter Brocklehurst, Director of BCTU, for his support for the trial. We are grateful to our surgical colleagues in Birmingham: Ms Natasha E Khan, Mr Phil Botha, Professor David J Barron and Dr Adrian Crucean, and Leeds: Mr Osama Jaber, Mr Imre Kassai and Mr Guiseppe Pelella for their input to the evolution of the trial. We thank Dr Oliver Stumper and Dr Anna N Seale, Birmingham for their advice on ICON monitoring and echo imaging, respectively. We thank Matt Hill and Alicia Gill at BCTU for programming and statistical support, respectively, and Collette Spencer for setting up the study in Leeds. We are most grateful to Martina Ponsonby and the trustees of *Young at Heart* for their feedback on the study documents.

**Contributors** NED, MM and TJJ conceptualised the trial. NED, RB, KPM, JM, NJI, PK and TJJ designed the trial with additional critical input from RW, JS, CvD and MM. NED, RLW and NJI developed the statistical analysis. NED and WBD designed the metabolic phenotyping substudy. All authors contributed to writing of the paper.

**Funding** This work is supported by an Intermediate Clinical Research Fellowship from the British Heart Foundation [FS/15/49/31612] awarded to NED. Lay review of the parent information sheets and consent forms was funded by a Patient & Public Involvement Bursary Award [RDS/WM-1318] from the NIHR Research Design Service in the West Midlands. The Institute of Cardiovascular Sciences is supported by an Accelerator Award from the British Heart Foundation [AA/18/2/34218]. Phenome Centre Birmingham was constructed through a grant from the Medical Research Council in the UK [MR/M009157/1].

**Disclaimer** Neither the sponsor nor funders had any role in the design of this study and will not have any role during its execution, analyses, interpretation of the data or decision to submit the results for publication.

**Competing interests** None declared.

**Patient and public involvement** Patients and/or the public were involved in the design, or conduct, or reporting, or dissemination plans of this research. Refer to the Methods section for further details.

**Patient consent for publication** Not required.

**Provenance and peer review** Not commissioned; externally peer reviewed.

**ORCID iDs**
Nigel E Drury http://orcid.org/0000-0001-9012-6683
Paulus Kirchhof http://orcid.org/0000-0002-1881-0197

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
