## [Reviewer comments · BMJ Open]

ARTICLE DETAILS

TITLE (PROVISIONAL)	The Bilateral Remote Ischaemic Conditioning in Children (BRICC) trial: protocol for a two-centre, double-blind, randomised controlled trial in young children undergoing cardiac surgery
AUTHORS	Drury, Nigel; Bi, Rehana; Woolley, Rebecca; Stickley, John; Morris, Kevin; Montgomerie, James; van Doorn, Carin; Dunn, Warwick; Madhani, Melanie; Ives, Natalie; Kirchhof, Paulus; Jones, Timothy

VERSION 1 – REVIEW

REVIEWER	Dr Ela Chakkarapani University of Bristol UK
REVIEW RETURNED	23-Jul-2020

GENERAL COMMENTS	1] Page 6: What was the largest trial and did the authors approach the authors for the data to enable a comprehensive metaanalysis. 2] Page 6: Please provide the research question in the PICO format followed by aim and objective. 3] Page 7: Please move the last 4 sentences in the inclusion criteria over to the exclusion criteria section 4] Page 8: Please clarify: Who identifies the patient? Who sends or gives who the PIS to parents? Will the clinician getting the consent for the trial be involved in the surgery for the child? 5] Given that the age spans from 3 months to 3 years, how do the authors account for the maturity of the myocardium on the primary outcome of hs-Troponin ? 6] Where does the intervention occur to enable blinding of the team that is providing the clinical care and please clarify how the anaesthetist looking after the child when the intervention happens can be blinded? 7] Please provide data to support the threshold of 50mmHg above the systolic BP. 8] If the systolic arterial blood pressure level for intervention was measured after induction of anaesthesia , would the anaesthesia not impact the blood pressure? 9] Intervention fidelity is lacking: 1] Does the loss of pulse oximetry trace last for 5 min? 2] In circumstances where the distal trace is not lost, there should be objective measures of temperature and a defined threshold of capillary refill time. Otherwise this will lead to a highly variable non-replicable intervention. 10] Blinding is not clear. Will the drape cover the thighs as well? Where will be the operating surgeon, anaesthetist during the intervention? Will it be the same person in each centre that will be applying the intervention for the whole study period? How is the consistency of implementation of the intervention recorded?
---

	11] Given that isoflurane has been reported to have preconditioning effect on myocardium, how will the investigators account for the variability in the use of inhaled anaesthetic agents? 12] Page 13: analysis. Please add appropriate reference for first sentence. 13] How many children will be in the Tetralogy group and VSD group? In the PIS, it is indicated that difference between the cyanotic and acyanotic lesions will be measured, however this is not addressed in the statistical analysis. 14] Primary outcome was AUC, however the sample size calculation utilises mean values. 15] Analysis: Ref 21 indicates the Values of Troponin in control group in the order of 24 microgram /L. Please explain how the value of 350 microgram/L/h was obtained for control group in this study to estimate the sample size. 16] How will the missing data be addressed? 17] Would be useful to add the PPI data that supports the study design, recruitment strategy and the intervention. The PPI data given does not cover the above aspects of the trial.
--	---

VERSION 1 – AUTHOR RESPONSE

1] Page 6: What was the largest trial and did the authors approach the authors for the data to enable a comprehensive metaanalysis.

- Response: The largest trial was that conducted by McCrindle et al in Toronto [27] which we refer to as such at the start of the paragraph beginning on page 5. If ‘did the authors approach...’ refers to us, we did not approach Dr Brian McCrindle & colleagues for their data to enable a comprehensive meta-analysis as our paper is a trial protocol. If it refers to Dr Wen Tan & colleagues from Beijing, the authors of the meta-analysis, we do not know whether they approached them, we only make reference to their findings as published.

2] Page 6: Please provide the research question in the PICO format followed by aim and objective.

- Response: We have reconfigured the first sentence of the rationale section on page 6 into the PICO format.

3] Page 7: Please move the last 4 sentences in the inclusion criteria over to the exclusion criteria section

- Response: The inclusion criteria paragraph on page 7 contains 2 sentences. If the reviewer is referring to the last sentence, this is a clarification of what is meant by tetralogy of Fallot in this context, noting that morphological variants which are managed and coded differently, were not included rather than excluded. The inclusion and exclusion criteria as stated are as per the current protocol and this trial has been running since October 2016; it therefore is not appropriate to change the inclusion and exclusion criteria at this stage.

4] Page 8: Please clarify: Who identifies the patient? Who sends or gives who the PIS to parents? Will the clinician getting the consent for the trial be involved in the surgery for the child?

- Response: Potential participants are identified by the principal investigator (PI) at each site (ND,

CvD) and this has been added to the recruitment paragraph on page 8. The PIS is given/sent out either by the PI or the research nurse. The Consultant Surgeon obtaining consent may or may not be involved in the surgery for that child, as this depends on other factors such the day of admission.

5] Given that the age spans from 3 months to 3 years, how do the authors account for the maturity of the myocardium on the primary outcome of hs-Troponin ?

- Response: We will include age and type of congenital heart defect in pre-defined sub-group analyses for the primary outcome; in the UK, complete repair of tetralogy of Fallot and VSD closure are usually performed at similar ages. However, we do not directly account for the maturity of the myocardium in the analysis. It is not known at what age the myocardium changes from an immature phenotype to an adult-like phenotype and whether factors such as chronic cyanosis impact on this change; this may occur before 3 months of age or later in different patients and we therefore cannot use a clinical variable to account for the maturity of the myocardium in the analysis.

6] Where does the intervention occur to enable blinding of the team that is providing the clinical care and please clarify how the anaesthetist looking after the child when the intervention happens can be blinded?

- Response: The intervention occurs after induction of anaesthesia, either in the anaesthetic room or the operating theatre. The anaesthetist is present to look after the child, but the surgical team are not. As mentioned on page 10, 'Blinding will be maintained by covering the child with a surgical drape from the nipples downwards throughout the period of cuff application, intervention, and removal.' This prevents the anaesthetist from seeing whether the cuffs are applied to the thighs or to 'the dummy limb... placed between the patient's legs.' We have added clarification that the area covered by the drape includes all four limbs.

7] Please provide data to support the threshold of 50mmHg above the systolic BP.

- Response: There is no specific data to support a threshold of 'at least 50mmHg above systolic pressure'. The aim of the intervention is to interrupt arterial flow for the defined period. As we point out, a criticism of the McCrindle study was that 'manual inflation of the cuff to just 15mmHg above systolic pressure may have led to periods of subclinical reperfusion and abolition of any preconditioning response' (pages 5-6). We therefore use a 'pressure-controlled tourniquet system set to at least 50mmHg above systolic pressure' as a pragmatic method of achieving cessation of arterial flow that is not susceptible to drift in cuff pressure or minor fluctuations in systolic pressure.

8] If the systolic arterial blood pressure level for intervention was measured after induction of anaesthesia, would the anaesthesia not impact the blood pressure?

- Response: The intervention is performed 'after induction of anaesthesia but prior to sternotomy' (page 9). Anaesthesia invariably causes a reduction in systolic blood pressure and as the intervention is performed whilst the child is anaesthetised, the tourniquet pressure is set immediately prior to the intervention according to the pressure shown on the arterial line in real-time. We have amended the text to make it clearer that this occurs in real-time.

9] Intervention fidelity is lacking: 1] Does the loss of pulse oximetry trace last for 5 min? 2] In circumstances where the distal trace is not lost, there should be objective measures of temperature and a defined threshold of capillary refill time. Otherwise this will lead to a highly variable non-replicable intervention.

- Response: 1) Yes, the trace is lost for the duration of the 5 minute ischaemic period, as indicated on

page 9: 'Continual loss of arterial flow will be confirmed by distal pulse oximetry during each limb occlusion cycle, visible only to the person applying the intervention'.

2) To clarify, if the distal trace is not lost, 'the cuff will be tightened or the inflation pressure increased to achieve arterial occlusion' (page 9). The reviewer refers to the clinical assessment to determine whether there is loss of arterial flow which would occur only 'if pulse oximetry is not available...' This option is included as a failsafe to cover the rare circumstance in which, for example, the pulse oximeter is not working; to date in this trial, this has occurred in <2% of cases and therefore will not lead to 'highly variable non-replicable intervention'. To our knowledge, no other trial of remote ischaemic preconditioning using distal oxygen saturation monitoring has considered a mechanism for dealing with a temporary lack of saturation monitoring. As the trial has been running since October 2016, it is inappropriate to further define objective measures.

10] Blinding is not clear. Will the drape cover the thighs as well? Where will be the operating surgeon, anaesthetist during the intervention? Will it be the same person in each centre that will be applying the intervention for the whole study period? How is the consistency of implementation of the intervention recorded?

- Response: As per point 6, and defined on page 10, 'Blinding will be maintained by covering the child with a surgical drape from the nipples downwards throughout the period of cuff application, intervention, and removal.' So yes, the drape covers the child from above the nipples downwards, including the thighs. The anaesthetist is present to look after the child and the surgical team are not, but the drape maintains blinding of all theatre staff.

The intervention is primarily applied by one person at each site; however, to cover for leave, unavailability or other absences, two additional staff at each site who are 'trained and competent in delivering the trial intervention, will perform the randomisation and administer the allocated treatment according to a standard operating procedure' (page 9).

11] Given that isoflurane has been reported to have preconditioning effect on myocardium, how will the investigators account for the variability in the use of inhaled anaesthetic agents?

- Response: As noted on page 10, '...end-tidal partial pressure [of isoflurane] will be recorded at the end of RIPC administration' and this will be reported for each group to determine any variability in the use of isoflurane. It will not be independently accounted for in the analysis, other than any variation by site.

12] Page 13: analysis. Please add appropriate reference for first sentence.

- Response: This is our hypothesis for this trial, it is not replicated from another study in the literature so there is no reference.

13] How many children will be in the Tetralogy group and VSD group? In the PIS, it is indicated that difference between the cyanotic and acyanotic lesions will be measured, however this is not addressed in the statistical analysis.

- Response: We are seeking to recruit similar numbers of patients with tetralogy of Fallot and VSD. In the statistical analysis section on page 14, we do state that: 'Test of interactions will be employed to assess whether there is evidence that the treatment effect differs between cyanotic and acyanotic patients.'

14] Primary outcome was AUC, however the sample size calculation utilises mean values.

- Response: These are mean values for area under the curve (AUC).

15] Analysis: Ref 21 indicates the Values of Troponin in control group in the order of 24 microgram /L. Please explain how the value of 350 microgram/L/h was obtained for control group in this study to estimate the sample size.

- Response: There were limited data in the paediatric literature on which to base our sample size calculation. The paper by Michael Cheung and colleagues published in J Am Coll Cardiol in 2006 (reference 21) was the best study on which to base our sample size calculation as it had a 'similarly mixed cohort of hypoxic and non-hypoxic children' (page 13) over 80% of whom had either tetralogy of Fallot or VSD. However, it does not contain numerical data for the values of troponin in the treatment or control groups. We therefore calculated the area under the curve for the two groups from the plot in figure 1 in their paper, hence the values of 350 for the control group and 228 for the treatment group, and the units of mcg/L/h. We believe the reviewer refers to the peak value at 6 hours in their control group from figure 1 which is approximately 22mcg/L.

16] How will the missing data be addressed?

- Response: As noted on page 14, 'A detailed statistical analysis plan is under development and will be approved prior to database lock' and this will include a framework for dealing with missing baseline and postoperative troponin data. Details of this will be included in the final results paper.

17] Would be useful to add the PPI data that supports the study design, recruitment strategy and the intervention. The PPI data given does not cover the above aspects of the trial.

- Response: As described on page 17, we sought PPI input to the study documents, the parent information leaflet and consent form, from parents and the Young Person's Steering Group. We did not directly seek PPI input to the study design, recruitment strategy and intervention so such data cannot be included in this protocol paper.

VERSION 2 – REVIEW

REVIEWER	Ela Chakkarapani St Michael's Hospital Translational Health Sciences University of Bristol
REVIEW RETURNED	30-Aug-2020
GENERAL COMMENTS	None